



**Electrical conductivity in the mantle transition zone beneath Mongol-**
**Okhotsk suture: revealed by the geomagnetic signals of ground**
**observatories**
Yanhui Zhang[(1)], Yuyan Zhang[(1)] , Longshuang Ma[(1)], Yue Yang[(2*)]
[1]Scool of Safety Engineering and Emergency Management, Shijiazhuang Tiedao University,
Shijiazhuang, 050043, China
[2]First Institute of Oceanography, Ministry of Natural Resources, Qingdao, 266061, China
*Correspondence: jlyangyue@126.com
**Abstract**:The closure of the Mongo-Okhotsk ocean has a strong influence on the
tectonic evolution of Northeast China. However, the dynamic mechanism in the
Mongol-Okhotsk suture area is controversial. This paper intends to obtain the deep
structure of beneath Northeast China based on geomagnetic depth sounding, and
constrain the subduction of Mongol-Okhotsk Ocean from the perspective of electrical
properties. This paper collects and processes the data of geomagnetic stations in China
and adjacent areas, and obtains stable *C*-response data. The staggered grid finite
difference method is used for forward modeling, and the finite memory quasi Newton
method based on $L_1$-norm is used for inversion. The three-dimensional inversion of
geomagnetic data is carried out in spherical coordinates. The intensive model testing
stations can obtain high-resolution underground electrical structures. The measured
data show that there are obvious high conductivity anomalies in the mantle transition
zone in Northeast China, especially in the west of the Great Xing'an Range, showing
an area of high conductivity anomalies. Combined with the regional tectonic
background of the region, we speculate that the high conductivity anomaly body is
related to the southward subduction of the Mongol-Okhotsk Ocean. The Mongol-
Okhotsk Ocean subducted under the Eurasian plate at a small angle in the southward
direction. With the closing of the Okhotsk Ocean and the extension environment after
the termination of subduction, the subducted oceanic crust plate has been faulted and
depressed and partially stopped in the mantle transition zone.



**Keywords:** Geomagnetic depth sounding; Three-dimensional inversion; Electrical structure;
Mongol-Okhotsk suture; Subduction

## 1. Introduction

It is generally believed that the tectonic evolution of Northeast China, like that of
South China and North China, is affected and controlled by the westward subduction
of the Western Pacific plate. However, the characteristics of other Mesozoic Cenozoic
sedimentary basins and the development of volcanic rocks in Northeast China are
obviously different from those in East China. It is difficult to explain these phenomena
simply by controlling the activities of the ancient Pacific plate. Besides the subduction
of the Western Pacific plate, the reason for this difference may also be related to its
unique tectonic location.
Mongol-Okhotsk suture zone locates in Northeast China, which extends from
central Mongolia to the Okhotsk Sea. The existence of the Paleozoic and Mesozoic
Mongol-Okhotsk Sea can be clearly seen from the Mongol-Okhotsk suture. The now
extinct Mongol-Okhotsk Sea is an ocean that existed in the Paleozoic (542 – 251 Ma;
Gradstein et al., 2004) and Mesozoic (251 – 66 Ma), and is located between the Siberian
continental block in the north and Amuria and North China continental block in the
south. It is difficult to reconstruct the history, geometry and closure of this ocean due
to the lack of sufficient paleomagnetic data and the sudden termination of the dispersion
suture to the west. The time and manner of ocean closure are not clear, which has led
to several alternative reconstructions.
Evidence of subduction related magmatism has been found on both sides of the
Mongol-Okhotsk suture (Zorin, 1999), indicating that subduction may have occurred
below the Siberian and Amurian margins, resulting in ocean closure. The
paleomagnetic data of the study also shows that the sealing began in the west and ended
in the east due to the coincidence of the rotation poles of the late Permian, early Triassic
and late Jurassic. This is supported by intrusions and marine fossils found in the young
suture from west to east (Zhao et al., 1990; Zonenshain et al., 1990; Halim et al., 1998;



Tomurtogo et al., 2005).

Previous studies focused more on the subduction of the Mongol-Okhotsk Ocean

to the Siberian craton in its north. In recent years, with the development of relevant
research work, some scholars gradually realized that the closure of the eastern segment
of the Mongol-Okhotsk Ocean has a wide impact on the tectonic deformation and
sedimentary formation in Northeast China (Tang et al., 2015). Based on the deep
seismic reflection data and the geochemical data of magmatic rocks, many scholars
believe that the Mongol-Okhotsk Ocean once subducted to the south (southeast), and
its closure process has squeezed the entire Northeast China and even the North China
Craton.

However, whether the Mesozoic tectonic evolution in the northern part of the

Great Xing'an Range (GXAR) is controlled by the Mongolia Okhotsk tectonic domain,
or by the subduction of the ancient Pacific plate, or both, is a matter of intense debate
(Sun et al., 2013; Xu et al., 2013).

As a geophysical exploration method that can obtain the deep electrical structure

of the earth, the geomagnetic depth sounding (GDS) is expected to obtain stable
characteristics of the deep electrical structure of the region, and further explore the
evidence of the closure and the southward subduction of the Mongol-Okhotsk Ocean,
providing further constraints for solving the above controversial issues.

GDS method is a unique tool to obtain deep mantle conductivity by inverting $C$-

responses (Kelbert et al., 2009; Munch et al., 2017; Grayver et al., 2017). Particularly,
with the application of three-dimensional (3-D) global electromagnetic (EM) induction
inversion method (Egbert and Kelbert, 2012) based on the mature 3-D forward in the
spherical coordinate system (Uyeshima and Schultz, 2000), GDS can now be used to
obtain a conductive structure closer to the real earth, thereby playing a vital role in
examining the conductivity heterogeneities of the earth (Utada et al., 2009; Kelbert et
al., 2009; Kuvshinov, 2012; Semenov and Kuvshinov, 2012; Püthe et al., 2015; Koch
and Kuvshinov, 2015).

China has densely distributed geomagnetic observatories. However, the present



researches don't make full use of them, resulting in poor resolution of the electrical
structure beneath China. Therefore, the existing three-dimensional (3-D) electrical
conductivity models are insufficient for solving the above problems. Kelbert et al (2008)
pointed out that the increase of the number of observatories can effectively improve the
resolution of GDS.

Zhang et al (2020) proposed a data processing method which can effectively

improve the utilization rate of the geomagnetic data. In addition, Li et al (2020)
proposed a $L_1$-norm 3-D GDS inversion technology basing on the limited-memory
quasi-Newton method (L-BFGS) which can greatly suppress the impact of noise data.
All these provide a theoretical basis for obtaining high-precision 3-D electrical
conductivity models in China.

In this study, more than 150 geomagnetic observatories widely distributed in China

are collected, and the BIRRP software is applied for stable $C$-response estimation. After
that, 50 high-quality response curves are obtained in and around China area. Basing on
the $L_1$-norm 3-D inversion method, the high-precision 3-D electrical conductivity
model of Northeast China is obtained. Combined with seismological and geological
information, the existence of Mongol-Okhotsk Ocean and its subduction is provided
with electrical constraints. Finally, the geodynamic process is discussed.

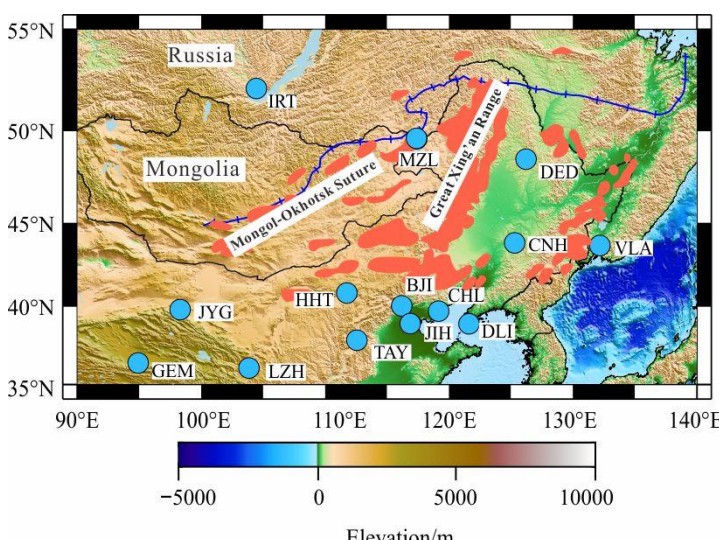




**Fig. 1.** Topographic map of present-day NE Asia with the locations of geomagnetic observatories and the Mongol-Okhotsk suture. Circles in blue represent the observatories, the distributions of igneous rocks are painted in orange.

## 2. Method

The inversion method applied in this paper is consistent with Li et al (2020), the $L_1$-norm is used to measure data misfit which is different with the normal inversion methods. $L_1$-norm can effectively curb the impact of the outliers which has been approved by Farquharson (2008), more details about the inversion method can be seen at Zhang and Yang (2022).

The forward $C$-response of the corresponding model in the process of inversion should be calculated numerically. As the basis of inversion, the selection of the forward modelling method is directly related to the accuracy of inversion results. The staggered-grid finite difference method is applied in this paper, since earth is a sphere, the forward method is used in a spherical coordinate system (Uyeshima and Schultz, 2000).

Different from the forward solver of Uyeshima and Schultz is the model gridding. In order to accelerate the inversion speed and try not to affect the accuracy of the inversion result, we use the local encryption method. In the research area, we will make the model mesh denser while the other region more sparsely (Li et al., 2020).

Most previous global electrical conductivity models are basing on the international shared geomagnetic observatories (about 11 observatories) in China (Kelbert et al., 2009; Li et al., 2020), which resulting in the low resolution of the earth models. This paper collects the densely distributed geomagnetic observatories in China and uses the data processing method based on BIRRP software (Zhang et al., 2020). Finally, 35 observatories can obtain high-quality $C$-response in and around China area.

The resolution of GDS inversion under such a dense distributed observatory has been tested in the previous research by Zhang and Yang (2022). The resolution tests show that the anomalies about 6° at the depth of mantle transition zone and the broken stagnant plate could be detected under a relative dense spread of the geomagnetic observatories. Therefore, 3-D inversion of the observatories in and around Mongol-





Okhotsk suture is expected to reveal the electrical structure of the mantle transition zone,
which may could provide useful information on constraining the evolutionary process
of the Mongol-Okhotsk ocean.

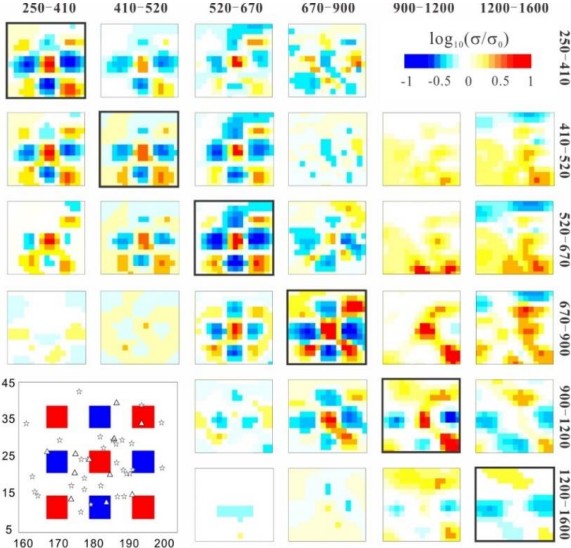


**Fig. 2.** Results of the resolution tests modified from Zhang and Yang (2022). The black boxes
are the locations of the designed anomalies; every column is an inversion result corresponding
a different layer of the anomalies, the amplitude of the anomalies is 10 times more conductive
or more resistive than the background model.

## 143    3. Real data inversion

Focus on the study area, 35 stable *C*-response curves of the observatories in

Northeast Asia (as shown in Fig. 1) are used for 3-D inversion. Their *C*-responses are
estimated based on the BIRRP software (Zhang et al. 2020). During inversion, the
background model is a 12-layer global mean conductivity model which is obtained by
Kelbert et al (2008). Since some observatories locate near ocean, the ocean effect must
be taken into account, therefore the surface conductance in the resolution of 1°× 1 °is
applied during the real data inversion.

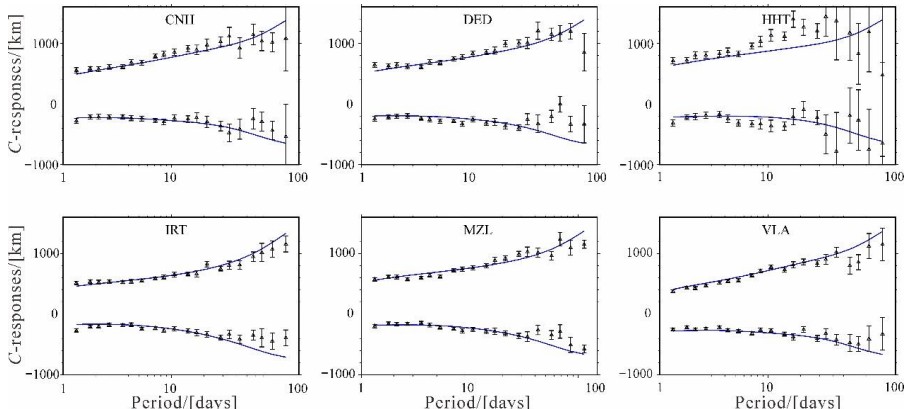

**Fig. 3.** Fitness curves of response of inversion results and the observed *C*-response for three geomagnetic observatories distributed around the Mongol-Okhotsk suture. blue lines are the inverted *C*-responses curves and the circles represent the observed *C*-responses.

After 79 iterations, our inversion terminates with an RMS (Root Mean Square) of 1.18. The RMS at most stations falls in the range of 1.1-1.3. About 80% of stations have RMS smaller than 1.4. The largest RMS is less than 1.7. This pattern of the RMS distribution suggests that the *C*-responses on all the stations can be fitted quite well, indicating that the inversion model is reliable. The fitting curves of typical observatories are shown in Fig. 3. It can be seen that the inverted *C*-responses fit well with the observed curves at most stations, especially at the short periods. As mentioned above, we think the inversion result is reliable.

The electrical structure beneath Northeastern China and its adjacent areas at layers of 410-520, 520-670, 670-900, 900-1200 km are plotted at Fig. 4. It can be seen that, in the layer of 520-660 km, there are two continuous low resistivity anomaly bodies distributed in a gourd shape along the Mongol-Okhotsk suture zone, and the scale of the anomaly body is about $10° \times 20°$, the average conductivity of the anomaly body is 2S/m, while the conductivity of the center of the anomaly body can reach 7S/m, about 7 times higher than the global average value (Kelbert et al., 2009). The above sensitivity tests can also explain the reliability of such large-scale electrical anomaly. To further verify the reliability of the abnormal, after fixing the electrical values of the grid model



in the abnormal area at 520-660 km depth, we conducted inversion again. The new
result showed that the RMS became larger, and high conductivity anomaly bodies
appeared at the boundary of the fixed area, which can also indicate the reliability of the
anomaly bodies in the target area.

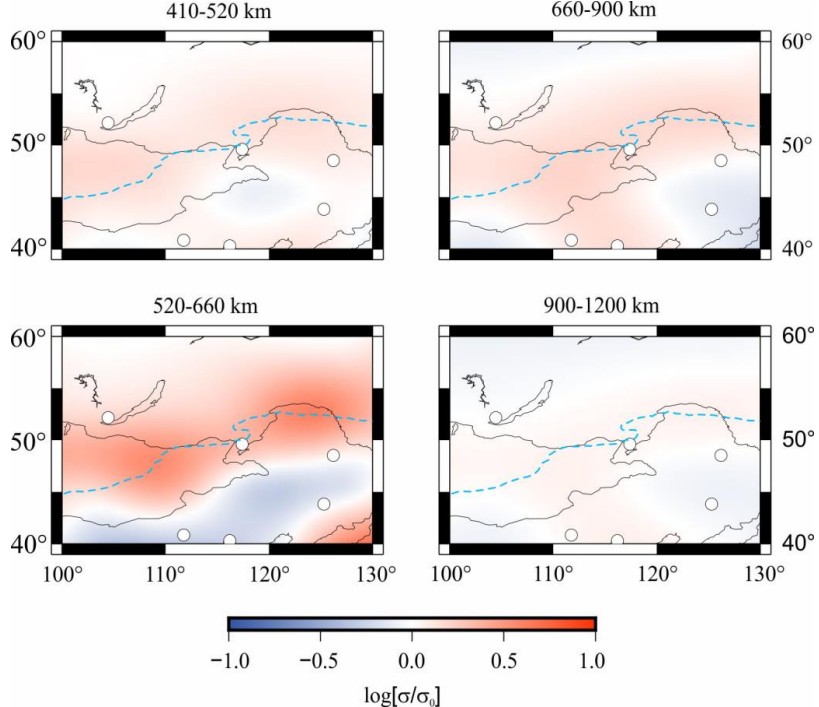


**Figure 4.** Slices of the electrical conductivity at the depth of 410-520, 520-670,
670-900, 900-1200 km of the inverted 3-D model.

## 4. Discussion

### 4.1 Subduction range of the Western Pacific plate

Seismic images show that the subduction plate of the Western Pacific plate lies flat
on the bottom of the mantle transition zone beneath East China, however, almost all the
seismic imaging results reveals that the subduction front of the Pacific plate seems to
be constrained to the east of the North-South Gravity Lineament (NSGL), and doesn't
exceed the NSGL (Ma et al., 2019; Zhao et al, 2009). Therefore, the high conductivity
abnormal seems to be independence with the subducted Pacific plate.



In addition, Yuan et al (2020) imaged the electrical resistivity structure in the depth
range of 350-1200 km beneath China by inverting the frequency-dependent ratios of
geomagnetic field component at a relatively dense network of geomagnetic
observatories. Their results also show that the western front of the subducting Pacific
plate in the MTZ roughly coincides with the abrupt change in the surface topography
in eastern China. Zhang et al (2020) obtained high-quality *C*-responses from the dense
geomagnetic observatories, and obtained the electrical structure of the MTZ of eastern
China. To the east of NSGL, most geomagnetic observatories show high conductivity
in MTZ; but for the observatories in the west of NSGL, the conductivity of most
observatories is relatively low. However, Manzhouli (MZL) observatory, which is far
away from NSGL shows obvious characteristics of high conductivity. This abnormal is
inferred to be caused by the mantle plume under Siberia craton. Due to the limitation
of 1-D inversion, it is difficult to show whether there is a connection between the high
conductivity beneath MZL observatory and the high conductivity found in MTZ of
eastern China.
Benefited from the realization of 3-D GDS inversion, several 3-D electrical
structure models of global or local regions have been obtained. Since the limitation of
the data density, most electrical models can only show that there is a high conductivity
anomaly model in MTZ beneath east china, the resolution of the anomaly body is
insufficient. Zhang et al (2022) used densely distributed observatories to obtain more
accurate inversion result. The new obtained result show that the stagnant Pacific plate
seems to be about 500 km away from NSGL. This may also indicate that the stagnant
plate in the MTZ of eastern China and the high conductivity anomaly body in the area
west of NSGL may have different formation mechanisms.
In summary, we concluded that the stagnant pacific plate could be constrained at
the east of NSGL, corresponding to the high conductivity abnormal at the east of
Songliao basin at the depth of 520-670 km as shown in Fig. 4. As for the high
conductivity beneath Mongol-Okhotsk suture, it seen to be less affected by the Pacific
tectonic domain.
**4.2 Sources of Volcanic materials in GXAR**



A NE trending volcanic rock belt with a length of 1700 km and a width of 900 km
is distributed in GXAR (Xu et al., 2013). Some researchers believe that magmatism in
GXAR is induced by mantle plume (Lin et al., 1999). However, there is no evidence of
earthquakes or He isotopes in the current research results indicating that there is a
mantle plume under the Xing'an-Mongolia Orogenic Belt (Huang and Zhao, 2006;
Chen et al., 2007). Isotopic dating has shown that the Early Cretaceous volcanic events
in GXAR lasted at least 30 Ma, which is inconsistent with the rapid manner in which
magma is formed in association with a mantle plume (Deng et al., 2019).
For the tectonic domain controlling the eruption, some scholars believe that it is
mainly related to the retreat of the Pacific plate (Faure and Natalin, 1992; Zhang et al.,
2010; Zhang et al., 2011; Ouyang et al., 2013, 2015). However, the temporal and spatial
distribution characteristics of volcanic rocks and the paleotectonic environment do not
support the above view (Engebretson et al., 1985; Maruyama and Seno, 1986; Kimura
et al., 1990; Yarmolyuk and Kovalenko, 2001). Xu et al. (2013) pointed out that the
Paleo Pacific tectonic domain mainly controls the magmatic and tectonic evolution in
the east of Songliao Basin.
In recent years, more and more people believe that the late Mesozoic magmatic
activity in the GXAR is related to the southward subduction of the Mongol-Okhotsk
Ocean (Zorin, 1999; Meng, 2003; Ying et al., 2010). In space, the eruption of Early
Cretaceous volcanic rocks in the northeast of GXAR was slightly later than that in the
northwest, which may be related to the gradual closing process of the Mongol-Okhotsk
Ocean from west to east (Cogné et al., 2005; Tomurtogoo et al., 2005; Metelkin et al.,
2010; Sun et al., 2013; Yang et al., 2015a)
In combination with the above geological analysis and the acquisition of high
conductivity anomaly bodies in the MTZ beneath the suture zone in this paper,. As
shown in Fig. 5, we propose a hypothesis that the Okhotsk Ocean subducted to the north
under the Siberian plate, and in the process of southward subduction, it subducted to
the lower part of Northeast China, and the oceanic crust materials in the two-way small
angle subduction entered the lower part of the land on both sides. Subsequently, the
Okhotsk Ocean gradually closed in a scissors style from west to east, and the Okhotsk



Ocean disappeared, making the Siberian Craton and the North China Craton compress
and collide with each other to form the Mongol-Okhotsk suture zone. The oceanic crust
materials brought in by subduction carry a lot of water into the mantle. Under the deep
thermal action, the subduction materials with high water content and asthenosphere
materials produce partial melting, and migrate upward due to buoyancy. During the
transmission to the top of the lithosphere, when encountering the dry old and hard
lithosphere in the upper part, they can only appear in the form of continental basalt
magma from tectonic weak units such as plate suture zones. Therefore, a large number
of igneous rocks are distributed in the western margin of Songliao Basin and the eastern
side of Mongolia Okhotsk suture zone. The oceanic crust material gradually sinks over
time. When it migrates to the MTZ, it is blocked by a 660 km discontinuous interface,
and the oceanic crust material stops at the lower layer of the mantle transition zone.

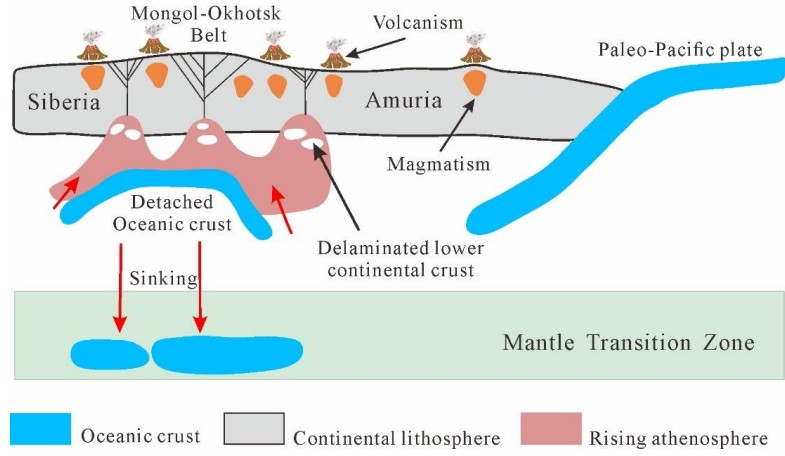


**Fig. 5.** Geodynamic model of the deep structure beneath Northeast Asia.

## 5. Conclusion


In this paper, based on the dense geomagnetic observatories net-work and the 3-D
GDS inversion technique, the high precision electrical structure beneath northeast
China was obtained. A high conductive abnormal beneath the Mongol-Okhotsk suture
was found, which may cause the widely spread igneous rocks. The main conclusions
drawn as a result of this study are as follows:



(1) High resolution electrical conductivity in the MTZ of the Northeastern China

was obtained basing on the dense geomagnetic data.

(2) The high conductor distributed in a gourd shape and parallel to the Mongol-

Okhotsk suture may be related to the closure of the Okhotsk ocean.

(3) The high conductor beneath Mongol-Okhotsk suture was speculated as the

subuducted oceanic crust materials, it leads the volcanic events on the ground

and sinking gradually at the bottom of the MTZ.

## Authorship Contribution Statement

Yanhui Zhang: Conceptualization, Methodology, Writing-review & editing.
Yuyan Zhang: Writing-original draft, Data curation. Yue Yang: Supervision,
Investigation, Writing-review & editing. Longshuang Ma: Visualization, Software,
Validation.

## Declare of Conflicting Interest

The authors declare that they have no known competing financial interests or
personal relationships that could have appeared to influence the work reported in this
paper.

## Acknowledgments

This work was supported by the National Natural Science Foundation of China
(Grant 42104079 and 42074080) and the Natural Science Foundation of Hebei Province
of China (Grant number D2021210007), to which we are very grateful. The authors
would like to thank National Geomagnetic Network Center of China for providing the
geomagnetic data. The inversion is developed on ModEM testbed. Some figures were
prepared using GMT software (Wessel & Smith, 1998).

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



446      56.

447