# Peer review of "Electrical conductivity in the mantle transition zone beneath Mongol- Okhotsk suture: revealed by the geomagnetic signals of ground observatories"

_EGUsphere, 2023_

## Referee Comment (RC2)

Electrical conductivity in the mantle transition zone beneath Mongol-Okhotsk suture: revealed by the geomagnetic signals of ground observatories

Review

This is an interesting paper that applies an existing technique to image the deep properties of the mantle transition zone.  The approach is not new, but it is applied to an area of China that has not been analysed before.  Overall, I think that the paper is worthy of being published but does need significant revision and clarification.

1.  The geophysical models and geological implications are highly dependent on the data set that has been developed.  I'd like to see much more about how the GDS responses were generated and what criteria were used in determining which sites were used in the 3D inversion.
2.  There is some confusing information about how many sites were used.  On line 98, the paper states that 150 sites were collected.  However, (on line 100) only 50 were used, but its not clear why the other 100 sites were rejected.  Figure 1 only shows 15 sites (blue dots), and later in the text (line 128 and line 144) the paper states that 35 were used.  Line 144 also states that these 35 sites are shown in Figure 1, but this is not the case.  Please clarify exactly what was used and why sites were excluded.
3.  The paper shows data from three sites (in Figure 3) which are described as "typical".  In my experience, "typical" usually means the best data sites with the most convincing model fits.  I'd like to see all the data shown as map images, maybe contoured slices for C responses at periods of 6 and 60 minutes.  This will allow the reader to really understand how homogeneous or heterogeneous the data are.  It might be useful to also show why 115 sites were not used in the inversion.
4.  The 15 sites shown in Figure 1 are also mostly clustered in the south but most of the inversion structure is located north of these sites.  In Figure 4, site IRT seems to constrain most of the structure, and if it were not included in the inversion the results might be very different.  I'm curious as to how sensitive the final model in Fig 4 is to the distribution of sites.
5.  It would have been good in Figure 4 to show the same area as in Figure 1.
6.  I'd like to have seen more information on the 3D inversion parameterisation.  Line 144 states that the ocean conductance was included in the model with a spatial resolution of 1° by 1°.  This is quite course (1° is about 100 km), and I'm curious to see how big the model is spatially.  I presume it is larger than the area shown in Figure 1, but it's not clear.
7.  I'm not really convinced by the resolution studies described Line 130-137 and Figure 2.  The chequer-board approach is fine in seismology, but for diffusive GDS methods the spatial resolution will always decrease with depth.  The chequer board model is far removed from any Earth structure in the mantle, so that inferences from the sensitivity tests are not useful.  I think this could be dropped, and instead its more important to show site coverage and data quality for the reader to understand the reliability of the model.
8.  The tectonic discussion is ok but is somewhat too detailed given that the only new evidence is the rather weakly constrained heterogeneity in the mantle transition zone.  I think it is reasonable for the authors to interpret the electrical models in terms of regional tectonic evolution, but Figure 5 is very speculative.  The observation of heterogeneity in the lower part of the mantle transition is an important one, and I think it would be better to spend more time discussing the mechanisms of enhancement.

Graham Heinson

University of Adelaide